# Incidence of Adnexal Torsion in the Republic of Korea: A Nationwide Serial Cross-Sectional Study (2009–2018)

**DOI:** 10.3390/jpm11080743

**Published:** 2021-07-29

**Authors:** Jin-Sung Yuk, Seung-Woo Yang, Myung-Hwa Lee, Min-Sun Kyung

**Affiliations:** 1Department of Obstetrics & Gynecology, Sanggye Paik Hospital, School of Medicine, Inje University, Seoul 01757, Korea; dryjs01@gmail.com (J.-S.Y.); mdmichaelyang@paik.ac.kr (S.-W.Y.); 2Department of Obstetrics & Gynecology, Hallym University Dongtan Sacred Heart Hospital, Hwaseong-si 18450, Kyeonggi-do, Korea; mhlee@hallym.or.kr

**Keywords:** adnexal torsion, incidence, age, follicular cyst, corpus luteal cyst, benign neoplasm

## Abstract

Background: This study aimed to evaluate the incidence and risk factors of adnexal torsion (AT) in the Korean population from 2009 to 2018 (10 years). Methods: We analyzed the 2009−2018 data obtained from the Health Insurance Review and Assessment Service National Inpatient Sample (HIRA-NIS) database. AT was identified by both diagnosis codes and surgery codes of adnexal surgery. Results: A total of 6,262,910 women were recorded in the database. The incidence of AT was 6 per 100,000 women (95% confidence interval (CI), 6−6). The incidence of AT tended to decrease with age after peaking in the late 20s to early 30s. In the weighted logistic regression analysis, women of childbearing age, especially in their 20s and 30s, had the highest AT incidence. Corpus luteal cyst (*p* < 0.001) and benign neoplasm (*p* < 0.001) increased the incidence of AT. Low socioeconomic status (SES), Charlson comorbidity index (CCI), and pregnancy were unrelated to AT. Conclusion: The incidence of AT is 6 per 100,000 women and peaks in the 20s to early 30s.

## 1. Introduction

Adnexal torsion (AT) is a disease in which the ligaments that support the ovaries and fallopian tubes are twisted, causing the blockage of some or all of the blood supply to the ovary [1,2,3]. Vascular obstruction can lead to tissue necrosis, which can be prevented through rapid treatment [4,5,6]. AT is responsible for approximately 3% and 2.7% of all surgical emergency diseases and gynecological emergency diseases, respectively [7]. Most of the previous studies on AT incidence rates have been based on data in a single-center patient population [7,8]. A study on the incidence of AT using National Health Insurance data has been previously reported in Korea; however, this investigation had limitations such as a small sample size and a short-term period of patient study, thus making it difficult to observe changes over time [8]. It also did not identify the associated risk factors. Therefore, the present study aimed to evaluate the incidence and risk factors of AT among Korean women using long-term data obtained from the National Health Insurance between 2009 and 2018 (10 years). To our knowledge, this study is the first to use 10-year data for analyzing AT incidence.

## 2. Materials and Methods

### 2.1. Data Collection

Almost all Koreans (approximately 98%) are required to have National Health Insurance [9]. According to the National Health Insurance Act, all medical institutions must treat patients covered by the National Health Insurance; then, they charge a portion of the treatment cost to the National Health Insurance Corporation. However, if the Health Insurance Corporation examines the adequacy of these expenses, medical institutions may suffer losses. Therefore, a neutral state agency called the Health Insurance Review and Assessment Service (HIRA) was established, specializing only in assessing cost adequacy [9]. HIRA shares most of its health insurance data with the National Health Insurance Corporation.

Furthermore, HIRA annually provides HIRA National Inpatient Samples (HIRA-NIS) for research purposes. The sample usually includes 10−13% (750,000 to 1 million) of patients with at least one hospitalization record and 1% (400,000) of patients without a hospitalization record every year. It is extracted using a stratified random sampling method. The collected data include most of the health insurance data (e.g., sex, age, treatment drug, surgical method, diagnosis code, hospital size, and insurance information), excluding treatment for cosmetic purposes and some drugs or surgeries that are not covered by the National Health Insurance. In this study, diagnosis codes and surgery codes were processed according to the International Statistical Classification of Diseases and Related Health Problems 10th edition (ICD-10) and Health Insurance Medical Care Expenses (2016 and 2018 versions). We analyzed HIRA-NIS data obtained from 2009 to 2018 (serial numbers: HIRA-NIS-2009-0066, HIRA-NIS-2010-0084, HIRA-NIS-2011-0163, HIRA-NIS-2012-0058, HIRA-NIS-2013-0085, HIRA-NIS-2014-0068, HIRA-NIS-2015-0057, HIRA-NIS-2016-0067, HIRA-NIS-2017-0015, and HIRA-NIS-2018-0050).

### 2.2. Patient Selection

AT was defined as the simultaneous presence of the AT diagnosis code (N83.5) and adnexal surgery code indicating that the surgery was performed.

### 2.3. Variables

In this study, the variables that might be related to AT occurrence included low socioeconomic status (SES), degree of comorbid disease, type of cyst (follicular cyst, corpus luteal cyst, or benign neoplasm of the ovary), and pregnancy. Low SES was considered when the medical insurance level was medical aid [10]. The Charlson comorbidity index (CCI) was calculated using the primary or sub-diagnosis of the patient’s concomitant disease in the calendar year. The presence of a diagnosis code related to pregnancy within 60 days of AT occurrence indicated that AT occurred during pregnancy. In the control group, pregnancy was defined when a diagnosis code related to pregnancy was present in the calendar year.

We searched for the presence of diagnosis codes for ovarian cysts/neoplasm in the same admission as the torsion. Among the follicular cyst (N83.0), corpus luteal cyst (N83.2), or benign neoplasm of the ovary (D27.x), at least one diagnosis code was defined as having the corresponding ovarian cyst/neoplasm. In the control group, if there was at least one ovarian diagnosis code for ovarian cysts/neoplasm in the calendar year, it was defined as having the ovarian cyst/neoplasm.

### 2.4. Statistics

All statistical data were analyzed using R 4.0.2 (The R Foundation for Statistical Computing, Vienna, Austria) and two-tailed test. A *p*-value of less than 0.05 was considered statistically significant. Categorical variables are presented as numbers (percentage), whereas continuous variables are presented as the mean ± standard deviation. In addition, non-normally distributed continuous variables are presented as the median [25th percentile; 75th percentile]. The categorical variables were analyzed using a weighted Chi-square test or weighted Fisher’s exact test, whereas the continuous variables were evaluated using a weighted *t*-test or weighted Mann–Whitney *U* test.

The incidence rate is presented as the rate per 100,000 people, using the weighted ratio. The risk for several variables was corrected by weighted logistic regression analysis. All missing values were processed with the mean imputation method.

### 2.5. Ethics

The Institutional Review Board of Hallym University Dongtan Sacred Heart Hospital approved this study (research number 2021-04-004).

Considering that this study used the HIRA sample data of inpatients, but removed inpatients’ personal identification variables, informed consent was not required from patients according to the Bioethics and Safety Act of Korea.

Although HIRA provided the data used for the analysis, the study results are not related to HIRA or the Korean Ministry of Health and Welfare.

## 3. Results

Out of the 6,262,910 patient data extracted from the HIRA-NIS database, 2890 were patients with adnexal torsion diagnosis code. Among them, a total of 1637 people were diagnosed with AT, excluding those without a surgery code (Figure 1). The median age of all patients was 34 [20; 51] years, but that of patients with AT was 30 [20; 41] years (Table 1). Table 1 lists the detailed characteristics of these patients.

The incidence rate of AT was 6 per 100,000 women (95% confidence interval (CI), 6−6). The most common surgical method for AT was oophorectomy or benign tumor removal, followed by adnexectomy, in our study. Table 2 lists the classifications of adnexal torsion-related surgical methods.

The incidence of AT tended to decrease with age after peaking in the late 20s to early 30s (Figure 2, Table 2). Figure 3 shows the incidence of AT according to year. Annual incidence rates of adnexal torsion were similar according to the year and were not significantly different over time.

In the weighted logistic regression analysis according to age per 5 years, SES, CCI, and pregnancy (Model 1), the incidence of AT decreased as the age increased at the 5-year-old interval (adjusted odds ratio (OR), 0.91; 95% CI, 0.88−0.94; *p* < 0.001) (Figure 4, Table 3). The low SES (adjusted OR, 1.58; 95% CI, 0.7−3.56; *p* = 0.268), CCI (CCI 1: adjusted OR, 0.67; 95% CI, 0.39−1.15; *p* = 0.144; CCI 2: adjusted OR, 0.67; 95% CI, 0.39−1.15; *p* = 0.144; CCI 3~: adjusted OR, 1.72; 95% CI, 0.94−3.17; *p* = 0.079), and pregnancy (adjusted OR, 1.66; 95% CI, 0.73−3.76; *p* = 0.228) were not associated with AT (Figure 4, Table 3). In the weighted logistic regression analysis of Model 2, the corpus luteal cyst (adjusted OR, 7.1;95% CI, 3.87−13.01; *p* < 0.001) and benign neoplasm of the ovary (adjusted OR, 110.81; 95% CI, 61.77–198.80; *p* < 0.001) increased the risk of developing AT (Figure 4, Table 3). In particular, the benign neoplasm of the ovary increased the risk of AT the most (Figure 4, Table 3). However, the follicular cyst was not associated with AT (adjust OR, 2.92; 95% CI, 0.84−10.17; *p* = 0.092) (Figure 4, Table 3).

## 4. Discussion

AT is an acute gynecological disorder with an incidence of 3% in a series of acute gynecological complaints [11,12,13].

The incidence among females aged 1 to 20 years in the United States was 4.9 per 100,000, analyzed using the Healthcare Cost and Utilization Project Kids’ inpatient database from 2000 to 2006 [14]. The AT incidence in pregnant women was 1.6 in 10,000 in the U.S. population using data from the Health Care Cost and Utilization Project Nation-wide Inpatient Sample from 2003 to 2011 [15]. However, few studies have reported the incidence of AT in large populations.

In this study, we analyzed the incidence of AT in a large national population from the HIRA-NIS database. Using the HIRA-NIS database, which includes long-term data (10 years) of a large population of Korean women (N = 6.8 million), sufficient and important information was obtained to analyze the incidence of rare AT.

Previously, there was an incidence report using HIRA-NIS data. Yuk et al. reported the incidence rate of adnexal torsion patients who underwent adnexal surgery using the HIRA-NIS database between 2009 and 2011. They reported that the incidence of torsion was 5.9 (95% CI, 5.4−6.4) per 100,000 people in Korea [8].

In our study, the incidence of AT was 6 per 100,000 women (95% CI, 6–6). The incidence of AT in our study was consistent with that in the study of Yuk et al. [8].

Furthermore, this study investigated AT incidence according to age. AT often occurs during reproductive ages, with 80% occurring before the age of 50 [16,17,18]. In our study, the incidence of AT peaked in the late 20s and early 30s and then decreased as the age increased. White et al. studied 52 women with surgery-confirmed AT. The median age was 33.5 years [28.7, 39.3] and the maximal incidence age was 30−34.9 years, consistent with our study results [18]. These results are related to the observed high incidence of ovarian masses occurring in women aged 21−40 years (112/212 (52.83%)) [19]. These results seem to be related to childbearing ages [17,18].

In the weighted logistic regression analysis, we confirmed that the corpus luteum cyst and benign neoplasm of the ovary are factors that increase the incidence of AT (*p* < 0.001). This result was consistent with the association between AT and ovarian mass in other studies [16,20]. In particular, benign neoplasm of ovary exhibited the highest OR related to AT. In another study, benign tumors accounted for 91.1% (31 out of 34) of AT cases [21]. As mentioned above, as a mechanism by which AT occurs frequently in benign neoplasms of the ovary, the increased size and weight of the involved ovary is thought to act as the fulcrum of the torsion [22]. Among benign neoplasms, AT is known to occur frequently in ovarian teratoma [19,23,24]. The reason for this high incidence is thought to be the weight and high fat contents of cyst, which induce “flotation” outside the pelvis, increasing the incidence of torsion [25,26]. Considering that teratoma occurs frequently at the childbearing age, this may explain why AT also occurs frequently at the childbearing age [27]. However, the insurance data we used do not have an independent diagnosis code for teratoma; thus, the relationship between teratoma and AT remains undetermined.

Hasson et al. reported that the presentation of adnexal torsion is similar in pregnant and nonpregnant women [28]. Moreover, in our study, pregnancy was not associated with the incidence of AT.

In a study by Yuk et al., pregnancy was reported to reduce the incidence of AT [29]. However, the study was a 1:4 matching study, and ovarian hyperstimulation syndrome was used as a risk correction variable. In our study, benign neoplasm and the CCI were corrected—an analysis technique that is different from that used in the study of Yuk et al. Therefore, directly comparing the two studies is not appropriate.

The occurrence of AT decreases when the ovary is not movable, as in endometriosis, tubo-ovarian abscess, and malignancy [18,20,30]. Endometriomas and malignant lesions that are associated with adhesions are relatively rare causes of AT, with malignant lesions accounting for approximately 2% of torsion cases [20]. However, in the present study, the association between AT and diseases with possible fixation, such as endometriosis, was not investigated [18,20,30].

Although the missing value was explained in the method, there was no missing value in our data. This is assumed to be because of two possible reasons. First, as HIRA data are insurance data, they are important in determining whether to pay insurance premiums. Therefore, it is thought that there is no missing value for the core code as the management of the core data is performed at the national level. Second, the NIS data we received are the sample data that were first processed by HIRA. Therefore, if there are missing values in the core code (age, gender, region, or insurance data) in HIRA, it seems to have been deleted.

This study has some limitations, as it is a study based on diagnosis codes. First, if the diagnosis code is incorrectly written before the operation, an error might exist. Additionally, if unexpected torsion is found during surgery in asymptomatic patients, the postoperative AT diagnosis code may be missing. However, surgery is a very sensitive issue in insurance claims, thus error occurrence is very unlikely when processing the surgery code. Therefore, the diagnosis code entered together with the surgery code will be highly accurate. Second, there are patients who have only a torsion diagnosis code and have not undergone surgery. Among them, there may actually be patients who initially had adnexal torsion and then spontaneously detorsioned. However, there is no way to accurately identify them without surgery. Third, there may be detection bias in this study, especially with regard to ovarian cysts and neoplasms. Benign ovarian cysts are common and may be less likely to be diagnosed in women unless symptoms are present. However, if a woman has adnexal torsion, then the process of investigation and treatment would lead to the detection and diagnosis of an ovarian cyst or neoplasm. Thus, it is possible that adnexal torsion leads to a bias in terms of increased detection/diagnosis of benign ovarian cysts and neoplasm.

This study provides important and useful data for identifying the incidence of AT in Koreans using HIRA-NIS, which has large-scale population data covering roughly 98% of Koreans.

## 5. Conclusions

The incidence of AT is 6 per 100,000 women. AT commonly occurs at the childbearing age, especially in the 20s to 30s. In addition, corpus luteum cyst and benign neoplasm of the ovary are factors that might increase the incidence of AT. These data help to us understand the incidence of AT, which is a rare reproductive condition. However, further research on a large scale is needed to identify more risk factors.

## Figures and Tables

**Figure 1 jpm-11-00743-f001:**
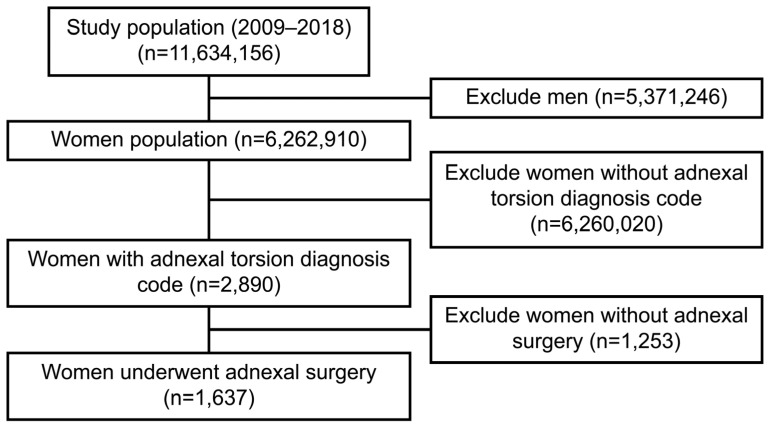
Flowchart for selecting women with adnexal torsion from the HIRA-NIS (2009–2018). HIRA, Korea Health Insurance Review and Assessment Service; NIS, national inpatient sample.

**Figure 2 jpm-11-00743-f002:**
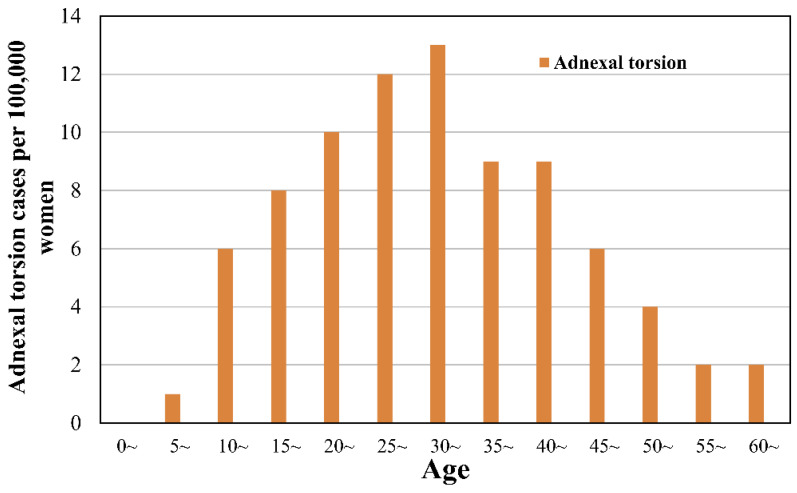
Incidence of adnexal torsion according to age at 5-year intervals (per 100,000 women) (HIRA-NIS 2009–2018).

**Figure 3 jpm-11-00743-f003:**
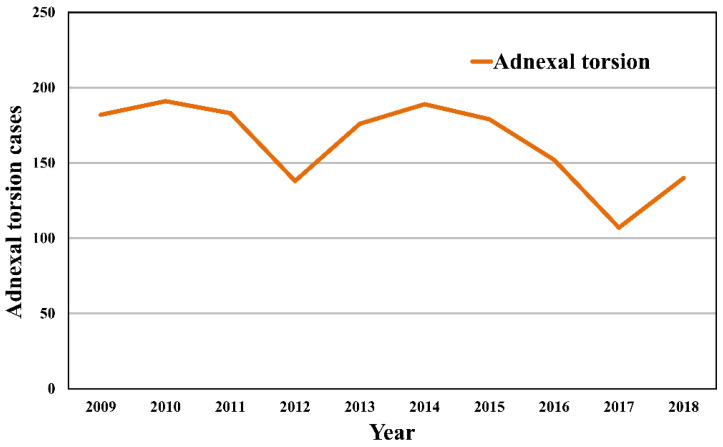
Adnexal torsion cases according to calendar year (HIRA-NIS 2009–2018).

**Figure 4 jpm-11-00743-f004:**
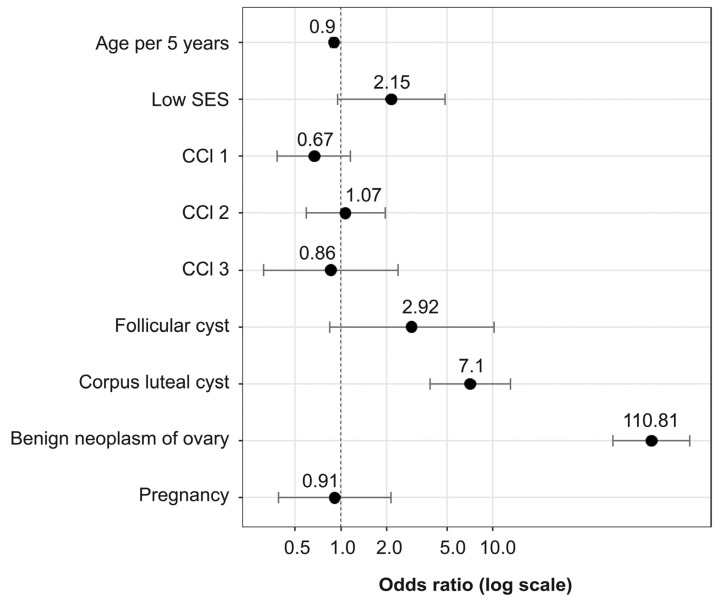
Odds ratio of adnexal torsion patients (HIRA-NIS 2018). HIRA-NIS, Health Insurance Review and Assessment Service national inpatient sample (in South Korea); SES, socioeconomic status; CCI, Charlson comorbidity index.

**Table 1 jpm-11-00743-t001:** Characteristics of women with AT in HIRA-NIS 2009–2018.

	AT			
	Non-AT (N = 6,261,273)	AT (N = 1637)	Total (N = 6,269,910)	*p*
Age (years)	34 [20;51]	30 [20;41]	34 [20;51]	<0.001 ^a^
Low SES				0.312
Absent	5,925,197 (94.6)	1570 (95.9)	5,926,767 (94.6)	
Present	336,076 (5.4)	67 (4.1)	336,143 (5.4)	
CCI				0.026
0	3,887,857 (62.1)	1245 (76.1)	3,889,102 (62.1)	
1	1,133,071 (18.1)	224 (13.7)	1,133,295 (18.1)	
2	730,774 (11.7)	118 (7.2)	730,892 (11.7)	
3~	509,571 (8.1)	50 (3.1)	509,621 (8.1)	
Follicular cyst				<0.001
Absent	6,254,427 (99.9)	1588 (97)	6,256,015 (99.9)	
Present	6846 (0.1)	49 (3)	6895 (0.1)	
Corpus luteal cyst				<0.001
Absent	6,211,163 (99.2)	1153 (70.4)	6,212,316 (99.2)	
Present	50,110 (0.8)	484 (29.6)	50,594 (0.8)	
Benign neoplasm of ovary				<0.001
Absent	6,190,807 (98.9)	760 (46.4)	6,191,567 (98.9)	
Present	70,466 (1.1)	877 (53.6)	71,343 (1.1)	
Right	21,481 (0.3)	353 (21.6)	21,834 (0.3)	
Left	19,523 (0.3)	224 (13.7)	19,747 (0.3)	
Unspecified	26,528 (0.4)	245 (15)	26,773 (0.4)	
Pregnancy				<0.001
Absent	5,671,790 (90.6)	1492 (91.1)	5,673,282 (90.6)	
Present	589,483 (9.4)	145 (8.9)	589,628 (9.4)	

Data are expressed as number (%) or median [25th percentile; 75th percentile]. AT, adnexal torsion; HIRA-NIS, Health Insurance Review and Assessment Service national inpatient sample (in South Korea); *p*, *p*-value; SES, socioeconomic status; CCI, Charlson comorbidity index. ^a^ Weighted Mann–Whitney *U* test was utilized.

**Table 2 jpm-11-00743-t002:** Adnexal torsion according to age per 5 years and surgery type (HIRA-NIS 2009–2018).

Age	Adnexectomy	Oophorectomy or Benign Tumor Removal	Oophorectomy of Malignant Tumor	Ovarian Wedge Resection	Vaginal Drainage of Ovarian Cyst	Total	Adnexal Torsion per 100,000 Women
	(N = 290)	(N = 1284)	(N = 8)	(N = 41)	(N = 14)	(N = 1637)	
≤4	1 (0.3%)	2 (0.2%)	0 (0%)	0 (0%)	0 (0%)	3 (0.2%)	0 (0–1)
5–9	1 (0.3%)	14 (1.1%)	0 (0%)	0 (0%)	0 (0%)	15 (0.9%)	1 (1–2)
10–14	11 (3.8%)	71 (5.5%)	2 (25%)	3 (7.3%)	0 (0%)	87 (5.3%)	6 (5–7)
15–19	10 (3.4%)	111 (8.6%)	2 (25%)	4 (9.8%)	3 (21.4%)	130 (7.9%)	8 (7–9)
20–24	18 (6.2%)	148 (11.5%)	0 (0%)	4 (9.8%)	0 (0%)	170 (10.4%)	10 (9–12)
25–29	31 (10.7%)	178 (13.9%)	2 (25%)	8 (19.5%)	1 (7.1%)	220 (13.4%)	12 (10–14)
30–34	29 (10%)	222 (17.3%)	0 (0%)	12 (29.3%)	4 (28.6%)	267 (16.3%)	13 (11–14)
35–39	42 (14.5%)	137 (10.7%)	1 (12.5%)	6 (14.6%)	2 (14.3%)	188 (11.5%)	9 (7–10)
40–44	55 (19%)	136 (10.6%)	0 (0%)	2 (4.9%)	1 (7.1%)	194 (11.9%)	9 (7–10)
45–49	38 (13.1%)	94 (7.3%)	0 (0%)	1 (2.4%)	2 (14.3%)	135 (8.2%)	6 (5–7)
50–54	26 (9%)	71 (5.5%)	0 (0%)	1 (2.4%)	0 (0%)	98 (6%)	4 (3–5)
55–59	8 (2.8%)	31 (2.4%)	0 (0%)	0 (0%)	0 (0%)	39 (2.4%)	2 (1–3)
60–64	7 (2.4%)	21 (1.6%)	1 (12.5%)	0 (0%)	0 (0%)	29 (1.8%)	2 (1–3)
65–69	4 (1.4%)	14 (1.1%)	0 (0%)	0 (0%)	0 (0%)	18 (1.1%)	1 (1–2)
70–74	3 (1%)	16 (1.2%)	0 (0%)	0 (0%)	1 (7.1%)	20 (1.2%)	2 (1–3)
≥75	6 (2.1%)	18 (1.4%)	0 (0%)	0 (0%)	0 (0%)	24 (1.5%)	1 (1–2)

Data are expressed as the number (%). AT, adnexal torsion; HIRA-NIS, Health Insurance Review and Assessment Service national inpatient sample (in South Korea).

**Table 3 jpm-11-00743-t003:** Logistic regression analysis of patients with AT (HIRA-NIS 2018).

	Adnexal Torsion			
	Unadjusted OR (95% CI)	*p*	Adjusted OR (95% CI)	*p*
Model 1 ^a^				
Age per 5 years	0.922 (0.879–0.948)	<0.001	0.91 (0.88–0.94)	<0.001
Low SES	1.337 (0.59–3.029)	0.487	1.58 (0.7–3.56)	0.268
CCI				
1	0.636 (0.371–1.092)	0.101	0.67 (0.39–1.15)	0.144
2	1.18 (0.663–2.098)	0.574	0.67 (0.39–1.15)	0.144
3~	0.752 (0.277–2.04)	0.575	1.72 (0.94–3.17)	0.079
Pregnancy	1.834 (0.809–4.155)	0.146	1.66 (0.73–3.76)	0.228
Model 2 ^b^				
Age per 5 years	0.922 (0.879–0.948)	<0.001	0.9 (0.85–0.94)	<0.001
Low SES	1.337 (0.59–3.029)	0.487	2.15 (0.95–4.83)	0.066
CCI				
1	0.636 (0.371–1.092)	0.101	0.67 (0.38–1.16)	0.149
2	1.18 (0.663–2.098)	0.574	1.07 (0.59–1.96)	0.826
3~	0.752 (0.277–2.04)	0.575	0.86 (0.31–2.37)	0.772
Follicular cyst	58.922 (21.573–160.932)	<0.001	2.92 (0.84–10.17)	0.092
Corpus luteal cyst	101.877 (71.53–145.1)	<0.001	7.1 (3.87–13.01)	<0.001
Benign neoplasm of ovary	202.915 (144.76–284.434)	<0.001	110.81 (61.77–198.80)	<0.001
Pregnancy	1.834 (0.809–4.155)	0.146	0.91 (0.39–2.14)	0.828

AT, adnexal torsion; HIRA-NIS, Health Insurance Review and Assessment service national inpatient sample (in South Korea); OR, odds ratio; CI, confidence interval; SES, socioeconomic status; CCI, Charlson comorbidity index. ^a^ Adnexal torsion = Age per 5 years + Low SES + CCI + pregnancy. ^b^ Adnexal torsion = Age per 5 years + Low SES + CCI + follicular cyst + corpus luteal cyst + benign neoplasm of ovary + pregnancy.

## Data Availability

The data presented in this study are available on request from the corresponding author.

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
