# Peer review of "Incidence of Adnexal Torsion in the Republic of Korea: A Nationwide Serial Cross-Sectional Study (2009–2018)"

_jpm, 2021, doi:10.3390/jpm11080743_

Round 1
Reviewer 1 Report
General comments
This manuscript describes a serial cross-sectional study of adnexal torsion in women in the Republic of Korea, using random samples of national health insurance data covering a 10-year period. The authors calculate the incidence of adnexal torsion, and investigate associations with age, calendar year, socioeconomic status, comorbidity, pregnancy, benign neoplasms of the ovary, and specific types of ovarian cyst.
Overall, I thought this was a good paper. The study investigates risk factors/ associations for a relatively rare but clinically important acute gynaecological condition, on which there is limited existing evidence. The use of national population data means that the results should be representative (at least of similar populations), and the large overall sample size enables reasonable statistical power even with a fairly rare condition.
I have some specific comments and suggestions for improvement (detailed below), but I think these should all be relatively easy for the authors to deal with.
Specific comments
- Title: I wonder if ‘serial cross-sectional study’ might be clearer than ‘cross-sectional study’? The authors have obtained a random sample of the health insurance data for each year over a period of 10 years, and so have effectively combined multiple cross-sectional studies.
- Introduction (page 1, lines 32-34): I think some of this sentence may be accidentally repetitive. [“A study on the incidence of adnexal torsion using National Health Insurance data has been previously reported in Korea was previously investigated using the National Health Insurance data…”].
- Materials and Methods (page 2, lines 69-70): “AT was defined as a case in which the AT diagnosis code (N.83.5) was present more than once in the main diagnosis or subdiagnosis.” Is this correct? It may be – but I wonder if the authors perhaps meant “at least once” rather than “more than once”?
- Materials and Methods (page 2, lines 83-84): I find the description of the definition of coexistent ovarian cysts and neoplasms a bit confusing. [“If each diagnosis code was found more than once, follicular cyst (N83.0), corpus luteal cyst (N83.2) or benign neoplasm of the ovary (D27.x) was considered.”] Could the authors perhaps explain a bit more? Did they look for the existence of these codes for ovarian cysts/ neoplasms just in the same admission as the torsion? What about for controls (the group without torsion) – did the authors just look for the presence of the codes for ovarian cysts/ neoplasms anywhere in that year?
- Materials and Methods (page 3, line 96): The authors state “All missing values were processed with the mean imputation method.” Could the authors perhaps give a bit more information regarding missing values (either here or in the Results)? The mean imputation method has some limitations (compared to, for example, multiple imputation), but this may not matter too much if there was relatively little missing data. On the other hand, it might have an impact if there is substantial missing data. It would therefore be helpful to have some more information (for example, what percentage of information was missing for certain key variables).
- Results (Table 1, page 3, line 117 onwards): I was unclear why the Total number of participants listed here in the top line of the table (N=6,283,788 in columns 4 and 8) does not match the total reported elsewhere (N=6,262,910). I think 6,262,910 would match the numbers quoted in the other columns better (e.g. 6,260,020 + 2,890 = 6,262,910).
- Results (Table 1, page 4, Pregnancy section, ‘Strict AT’ section, columns 6-8): I think the percentages quoted here may be wrong (e.g. 22.6, 2.3; 19.4, 1.9; 22.6, 2.3). The rest of the table quotes column percentages for each variable, with each column adding to 100% (e.g. for pregnancy in columns 2-4: 90.6, 9.4; 89.8, 10.2; 90.6, 9.4).
- Results (Table 3, page 7): The authors refer to ‘Formula 1’ and ‘Formula 2’. While I understand what they mean, I think some readers might find it easier if they referred to them as ‘Model 1’ and ‘Model 2’.
- Discussion (page 8, line 219-220): I think there may be an incomplete sentence here – “In the present study, the association between AT and diseases with possible fixation, such as endometriosis,[18,20,30].” As far as I could see, conditions such as endometriosis were not investigated in this study?
- Discussion: I think it might be helpful if the authors discussed the possibility of detection bias in this study, particularly with regard to ovarian cysts and neoplasms. Benign ovarian cysts are common, and it seems likely that in many women they will never be diagnosed, so long as the women are asymptomatic. However, if a woman has adnexal torsion, then the process of investigation and treatment would lead to the detection and diagnosis of any ovarian cyst or neoplasm. Thus, it is possible (arguably likely) that adnexal torsion leads to a bias in terms of increased detection/ diagnosis of benign ovarian cysts and neoplasms. I would therefore be cautious about statements such as “In addition, corpus luteal cyst and benign neoplasm increased the incidence of AT.” [Conclusions, page 9, lines 238-239]. I think it is true to say that diagnoses of corpus luteal cysts and benign neoplasms were more common in women who had a diagnosis of adnexal torsion than in women without adnexal torsion. Part of this association may well be causal (i.e. that corpus luteal cysts and benign neoplasms increase the risk of developing adnexal torsion), but some of the association is also likely to be due to detection bias (i.e. that adnexal torsion increases the chances of detecting a corpus luteal cyst or benign neoplasm if it is there).
Reviewer 2 Report
The authors aimed to report the of adnwxal torsion in the rupublic of Korea in a 10 years period
I would appreciate if the authors could comment on the following remarks:
- could you wxplain to differentiate the AT to AT and stict AT . Actually the non stict AT ate confusing and their relevance is questionable and misleading. Should not AT torsion diagnosis be given only to operated ones?How certain can you be with out surgery?
- With respect to remark 1 there is no explanation at all in methods to way of diagnosis. Sonographic one? What were the clinical/sonographic signs to define the " non strict AT"
Round 2
Reviewer 2 Report
I thank the authors for their response.
The inclusion of non stict AT might be misleading. The explanation that detorsion spontaneously occured assumes or determines that torsion did exist and was the only explanation for the clinical appearance . The fact the clinicaly the patient is free of pain can not be diagnoses as clear torsion-detorsion.
I offer the authors to remove this section of non strict AT form the study and consentrate on the proved AT
